

# Newton series expansion of bosonic operator functions

**Jürgen König⋆ and Alfred Hucht**

Fakultät für Physik, Universität Duisburg-Essen and CENIDE,
D-47048 Duisburg, Germany

⋆ koenig@thp.uni-due.de

## Abstract

We show how series expansions of functions of bosonic number operators are naturally derived from finite-difference calculus. The scheme employs Newton series rather than Taylor series known from differential calculus, and also works in cases where the Taylor expansion fails. For a function of number operators, such an expansion is automatically normal ordered. Applied to the Holstein-Primakoff representation of spins, the scheme yields an exact series expansion with a finite number of terms and, in addition, allows for a systematic expansion of the spin operators that respects the spin commutation relations within a truncated part of the full Hilbert space. Furthermore, the Newton series expansion strongly facilitates the calculation of expectation values with respect to coherent states. As a third example, we show that factorial moments and factorial cumulants arising in the context of photon or electron counting are a natural consequence of Newton series expansions. Finally, we elucidate the connection between normal ordering, Taylor and Newton series by determining a corresponding integral transformation, which is related to the Mellin transform.

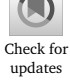

# 1 Introduction

Functions of operators are ubiquitous in physics. In this paper, we focus on functions $f$ that depend on the occupation number operator $\hat{n} = \hat{a}^{\dagger}\hat{a}$, where $\hat{a}^{\dagger}$ and $\hat{a}$ are the creation and annihilation operator of some bosonic mode in second quantization, with commutation relation $[\hat{a}, \hat{a}^{\dagger}] = 1$. For any (real or complex) function $f(x)$, the operator-valued function $f(\hat{n})$ is defined through the eigenvalue equation

$$f(\hat{n})|n\rangle = f(n)|n\rangle, \tag{1.1}$$

where $|n\rangle$ is an eigenstate of $\hat{n}$, with integer eigenvalue $n \in \mathbb{N}_0$. While for simple functions, such as $f(\hat{n}) = \hat{n}$ or $\hat{n}^2$, this definition may be sufficient in order to perform practical calculations, for more complicated functions, such as $f(\hat{n}) = \sqrt{\hat{n}}$, a series expansion of $f(\hat{n})$ in terms of $\hat{a}^{\dagger}$ and $\hat{a}$ may be desirable because, then, an approximative treatment of the problem at hand becomes possible by truncating the series at some low order.

One of the most prominent series expansion used in physics is the Taylor expansion (without loss of generality around $x = 0$),

$$f(x) = \sum_{k=0}^{\infty} \frac{1}{k!} \partial_x^k f(0)\, x^k, \tag{1.2}$$

with $\partial_x^k f(0)$ being the $k$-th derivative of $f(x)$ at $x = 0$, which is valid if $f(x)$ is analytic at the expansion point $x = 0$. In that case, $x^k$ can be replaced by $\hat{n}^k$ in (1.2) to obtain the formal power series of the operator function $f(\hat{n})$. And, indeed, such a procedure is commonly used, e. g., to expand spin operators in terms of Holstein-Primakoff bosons, as we discuss in more detail below.

It should be emphasized, however, that the choice of how to order the operators $\hat{a}$ and $\hat{a}^{\dagger}$ in the series expansion is not unique. While the Taylor expansion (1.2) yields products of the form $(\hat{a}^{\dagger}\hat{a})^k$, one may rearrange the operators in some other way. For example, rewriting with the help of $[\hat{a}, \hat{a}^{\dagger}] = 1$ the *second*-order term in normal order, $\hat{a}^{\dagger}\hat{a}\hat{a}^{\dagger}\hat{a} = \hat{a}^{\dagger}\hat{a}^{\dagger}\hat{a}\hat{a} + \hat{a}^{\dagger}\hat{a}$, modifies the coefficient of the *first*-order term. As a consequence, the series expansion of $f(\hat{n})$ is *not* unique and depends on the operator order convention.

An intrinsic feature of the Taylor expansion is that it requires $f(x)$ to be analytic at the expansion point. Therefore, the above procedure does *not* work, e. g., for the operator square root $\sqrt{\hat{n}}$ since, due to the divergence of the differential quotient $\mathrm{d}f/\mathrm{d}x$, the square root $\sqrt{x}$ is not analytic at $x = 0$. On the other hand, since we only need to consider the function $f(x)$ at *integer* values of $x$, diverging differential quotients should be irrelevant for the possibility to find a series expansion of $f(\hat{n})$.

This motivates us to suggest that, from the very beginning, the Taylor series known from differential calculus should be replaced by the Newton series, a central tool from finite-difference calculus. For the Newton series to exist, the only requirement is that $f(x)$ is well defined at integer values of $x$. This includes non-analytic functions such as $\sqrt{x}$, i. e., a series expansion of $\sqrt{\hat{n}}$ becomes possible. But also for analytic functions $f(x)$, for which the Taylor series exists, employing the Newton series to expand $f(\hat{n})$ seems more natural and better adapted to the discreteness of the domain of definition of $f(\hat{n})$, as will be detailed in the following.

# 2 Finite-difference calculus

Although the Newton series and the underlying finite-difference calculus was invented a long time ago [1, 2], it is, among physicists today, less known and used than the Taylor expansion

that has been invented later [3]. Therefore, we briefly review this expansion scheme before applying it to operator-valued functions of the form (1.1).

## 2.1 Newton series

In finite-difference calculus [4,5], the differential quotient is replaced by the forward difference

$$\Delta_n f(n) = f(n+1) - f(n). \tag{2.1}$$

Applying the difference operator $\Delta_n$ iteratively $k \geq 0$ times yields the $k$-th order forward difference

$$\Delta_n^k f(n) = \sum_{l=0}^{k} (-1)^{k-l} \binom{k}{l} f(n+l). \tag{2.2}$$

The discrete analog[1] of the Taylor series in differential calculus is the Newton series. It is, for an expansion starting at $n = 0$, given by

$$f(n) = \sum_{k=0}^{\infty} \frac{1}{k!} \Delta_n^k f(0) \, n^{(k)}, \tag{2.3}$$

where

$$n^{(k)} = n(n-1)(n-2)\cdots(n-k+1) \tag{2.4}$$

denotes the $k$-th (falling) factorial power[2,3] of $n$. Comparing equations (1.2) and (2.3), we point out that the factorial powers $n^{(k)}$ are the discrete analog of the usual powers $n^k$.

It should be emphasized that, by construction, the $r$-th partial sum (i.e. up to order $n^{(r)}$) of the Newton series *exactly* reproduces $f(n)$ at the integer values $n = 0, 1, \ldots, r$. In fact, the $r$-th partial sum is equal to the $r$-th order Lagrange interpolation polynomial through the $r+1$ points $(0, f(0)), \ldots, (r, f(r))$. Therefore, the Newton series converges pointwise at all $n \in \mathbb{N}_0$, with the only requirement that $f(n)$ is well defined on $\mathbb{N}_0$. While the nonanalyticity of $\sqrt{x}$ at $x = 0$ prevents the expansion into a Taylor series, there is no problem of expanding $\sqrt{n}$ into a Newton series, with the first few terms given by

$$\sqrt{n} = n - \frac{2 - \sqrt{2}}{2!} n^{(2)} + \frac{3 - 3\sqrt{2} + \sqrt{3}}{3!} n^{(3)} + \mathcal{O}\left(n^{(4)}\right). \tag{2.5}$$

## 2.2 Number operator functions

For the Newton expansion of the operator function $f(\hat{n})$, one simply has to replace $n$ with $\hat{n}$ in (2.3). This yields

$$f(\hat{n}) = \sum_{k=0}^{\infty} \frac{F_k}{k!} \hat{n}^{(k)}, \tag{2.6a}$$

---

[1]    Correspondences such as (1.2) vs. (2.3) are also investigated in the framework of the umbral calculus [6].

[2]    An analog discussion can be made based on backward differences $\nabla_n f(n) = f(n) - f(n-1)$ in combination with the rising factorial power.

[3]    There are different definitions and notations for the factorial power in the literature. We use the same definition as *Mathematica* [7], which obeys the relations $\Delta_n n^{(k)} = k n^{(k-1)}$ as well as $\sum_n n^{(k)} = n^{(k+1)}/(k+1)$ analog to differentiation and integration.

where the coefficients

$$F_k = \Delta_n^k f(0) = \sum_{l=0}^{k} (-1)^{k-l} \binom{k}{l} f(l) \tag{2.6b}$$

are also known as the binomial transform $F$ of $f$ [8].

An alternative and direct way to derive (2.6) uses the fact that, contrary to differentiation, finite differences can also be defined with respect to the *operator* $\hat{n}$ instead of the number $n$, because neither a division nor a limit is necessary in the definition of the forward difference operator

$$\Delta_{\hat{n}} f(\hat{n}) = f(\hat{n}+1) - f(\hat{n}), \tag{2.7}$$

and likewise for $\Delta_{\hat{n}}^k f(\hat{n})$. To formulate the operator Newton series, we need to take the matrix element of $\Delta_{\hat{n}}^k f(\hat{n})$ at the expansion point. For an expansion around the lower bound of the spectrum of $\hat{n}$, this leads to the coefficients $F_k = \langle 0 | \Delta_{\hat{n}}^k f(\hat{n}) | 0 \rangle$, which are identical to $\Delta_n^k f(0)$ by (1.1).

## 2.3 Factorial powers and normal ordering

The Newton series (2.6) of $f(\hat{n})$ contains the factorial powers $\hat{n}^{(k)} = \hat{n}(\hat{n}-1)\cdots(\hat{n}-k+1)$ of the number operator $\hat{n} = \hat{a}^{\dagger}\hat{a}$. These are known to be identical [9] to the normal-ordered (regular) powers of $\hat{n}$, i.e.,

$$\hat{n}^{(k)} = \, :\hat{n}^k: \, = \hat{a}^{\dagger k} \hat{a}^k. \tag{2.8}$$

The action of the normal-ordering operator $:\cdot:$ on any function $f(\hat{n})$ is calculated by first expressing $f(\hat{n})$ through its formal power series, then replacing each $\hat{n}$ by $\hat{a}^{\dagger}\hat{a}$, and finally shifting all the creation operators $\hat{a}^{\dagger}$ to the left and all the annihilation operators $\hat{a}$ to the right, while ignoring the non-commutativity of $\hat{a}$ and $\hat{a}^{\dagger}$. The result is, of course, different from using the commutator $[\hat{a}, \hat{a}^{\dagger}] = 1$ to rewrite $f(\hat{n})$ in a normal-ordered form.

Relation (2.8) is easily proven by applying it to a number eigenstate $|n\rangle$. For $k \leq n$, we get

$$:\hat{n}^k: |n\rangle = \hat{a}^{\dagger k} \hat{a}^k |n\rangle = \sqrt{n^{(k)}} \, \hat{a}^{\dagger k} |n-k\rangle = n^{(k)} |n\rangle = \hat{n}^{(k)} |n\rangle, \tag{2.9}$$

while for $k > n$ the relation $:\hat{n}^k: |n\rangle = \hat{n}^{(k)} |n\rangle$ is trivially fulfilled, as then both $\hat{a}^k |n\rangle = 0$ and $n^{(k)} = 0$.

By combining the results (2.6) and (2.8) from above, we conclude that the Newton series expansion of any function of number operators is *always* normal ordered,

$$f(\hat{n}) = \sum_{k=0}^{\infty} \frac{\Delta_n^k f(0)}{k!} \hat{a}^{\dagger k} \hat{a}^k. \tag{2.10}$$

The normal ordering of the series expansion (2.10) implies that the $k$-th addend does not contribute when applied to number eigenstates $|n\rangle$ with $n < k$. As an important consequence, the Newton series of a function $f(\hat{n})$ with a finite support $\{|n\rangle; n \leq r\}$ terminates with the $r$-th partial sum.

## 2.4 Finite differences and commutators

We remark that the difference operator $\Delta_{\hat{n}} f(\hat{n})$ from (2.7) is connected to the commutator[4]

$$[\hat{a}, f(\hat{n})] = \hat{a} f(\hat{n}) - f(\hat{n}) \hat{a} = f(\hat{n}+1) \hat{a} - f(\hat{n}) \hat{a} = \Delta_{\hat{n}} f(\hat{n}) \hat{a}, \tag{2.11}$$

---

[4] The corresponding relation involving the creation operator is $[f(\hat{n}), \hat{a}^{\dagger}] = \hat{a}^{\dagger} \Delta_{\hat{n}} f(\hat{n})$.

which resembles the commutator $[\hat{p}, f(\hat{x})] = \frac{\hbar}{i}\partial_x f(\hat{x})$ between the momentum operator and a *continuous* function of the position operator, involving the *derivative* of $f$. More generally, the $k$-th order difference operator $\Delta_{\hat{n}}^k f(\hat{n})$ is related to the $k$-fold commutator

$$[\hat{a}, f(\hat{n})]_k = \underbrace{[\hat{a}, [\ldots, [\hat{a}, [\hat{a}, f(\hat{n})]]]]}_{k-\text{fold}} = \Delta_{\hat{n}}^k f(\hat{n}) \hat{a}^k, \tag{2.12}$$

which is proven by iteratively applying the steps used in (2.11). These $k$-fold commutators can be derived from the generating function

$$\hat{G}(t) = e^{t\hat{a}} f(\hat{n}) e^{-t\hat{a}} \tag{2.13}$$

via $[\hat{a}, f(\hat{n})]_k = \partial_t^k \hat{G}(0)$. As a consequence of (2.12), the series expansion of the generating function in $t$ is given by

$$\hat{G}(t) = \sum_{k=0}^{\infty} \frac{t^k}{k!} \Delta_{\hat{n}}^k f(\hat{n}) \hat{a}^k, \tag{2.14}$$

which is not yet normal ordered. To achieve a normal-ordered representation, we additionally expand $f(\hat{n})$ into a Newton series (2.10) to finally arrive at

$$\hat{G}(t) = \sum_{k=0}^{\infty} \frac{t^k}{k!} \sum_{l=0}^{\infty} \frac{\Delta_n^{k+l} f(0)}{l!} \hat{a}^{\dagger l} \hat{a}^{l+k}. \tag{2.15}$$

In the following, we present three examples of Newton series expansions of bosonic operator functions from different fields of physics. While the first example illustrates the advantages of Newton series in the framework of the Holstein-Primakoff representation of quantum spins, the second example deals with coherent states and the related displacement operators. Finally, the third example considers many-particle quantum statistics and the relations to factorial moments.

## 3 Applications

### 3.1 Bosonic representation of spins

As a first example, we study the Holstein-Primakoff (HP) representation [10] of quantum spins of length $S$. It is given by (we put $\hbar = 1$)

$$\hat{S}^+ = \sqrt{2S}\, h(\hat{n})\, \hat{a}, \qquad \hat{S}^- = \sqrt{2S}\, \hat{a}^\dagger h(\hat{n}), \qquad \hat{S}^z = S - \hat{n}, \tag{3.1a}$$

with the bosonic operator function

$$h(\hat{n}) = \sqrt{1 - \frac{\hat{n}}{2S}}, \tag{3.1b}$$

and, thus, expresses the spin operators in terms of bosonic creation and annihilation operators $\hat{a}^\dagger$ and $\hat{a}$, respectively. The spin state $|S, m\rangle_{\text{spin}}$ with the largest magnetic quantum number $m = S$ is identified with the bosonic vacuum state $|0\rangle$. Creating one boson by applying $\hat{a}^\dagger$ reduces $m$ by one. The function $h(\hat{n})$ specified in (3.1b) guarantees that the commutation relations of the spin algebra,

$$[\hat{S}^+, \hat{S}^-] = 2\hat{S}^z \qquad \text{and} \qquad [\hat{S}^z, \hat{S}^\pm] = \pm\hat{S}^\pm, \tag{3.2}$$

are satisfied. As a consequence, the Holstein-Primakoff representation yields the correct matrix elements of the spin operators.

The Hilbert space of the HP bosons has infinite dimension and is, therefore, much larger than the $(2S+1)$-dimensional Hilbert space of the quantum spin. However, the relation $\hat{S}^-|S, -S\rangle_{\text{spin}} = 0$, responsible for keeping the spin Hilbert space finite-dimensional, properly translates to $\hat{S}^-|2S\rangle = 0$ in the bosonic language, because of $h(2S) = 0$. Therefore, the unphysical part of the boson Hilbert space, with more than $2S$ bosons present, can never be reached by applying the spin operators and therefore decouples from the physical part.

## Series expansions of Holstein-Primakoff representation

Since the square root in (3.1b) is awkward to deal with, a common procedure often used in the literature to analyze collective magnetic excitations (magnons) in ferromagnetic and antiferromagnetic spin lattices is to expand $h(\hat{n})$ from (3.1b) in a Taylor series up to $r$-th order, $h(\hat{n}) = h_{\text{T},r}(\hat{n}) + \mathcal{O}(\hat{n}^{r+1})$. To lowest order, this expansion yields $h_{\text{T},0} = 1$. This defines the so-called linear spin-wave approximation, which can be understood as the classical, or $S \to \infty$, limit of the quantum spin, in which interactions between magnons are neglected.

To address magnon-magnon interactions, at least the next order of the Taylor expansion needs to be included,

$$h(\hat{n}) = \underbrace{1 - \frac{\hat{n}}{4S}}_{h_{\text{T},1}(\hat{n})} + \mathcal{O}(\hat{n}^2). \tag{3.3}$$

The truncation of the Taylor series at any *finite* order $r > 0$ is, however, problematic since it makes the full bosonic Hilbert space accessible by successively applying $\hat{S}^-$, including all the unphysical states with more than $2S$ bosons. Furthermore, the canonical commutation relations for the spin operators are only satisfied approximately. This results in artificially breaking rotational symmetries that may be present in the original Hamiltonian. Only when the Taylor series is kept up to *infinite* order, both of these problems are cured.

Instead of using the Taylor expansion, we advocate to employ the Newton expansion (2.10) up to order $r$ and define $h(\hat{n}) = h_r(\hat{n}) + \mathcal{O}(\hat{n}^{(r+1)})$, which to lowest order yields the same result $h_0 = 1$. However, in next-to-leading order $r = 1$ they start to differ. The Newton expansion yields

$$h(\hat{n}) = \underbrace{1 - [1 - h(1)]\,\hat{n}}_{h_1(\hat{n})} + \mathcal{O}(\hat{n}^{(2)}), \tag{3.4}$$

with a different prefactor for the linear term than in the Taylor expansion[5]. As a consequence, the magnon-magnon interaction in spin lattices acquires a different strengths for the truncated Taylor and Newton expansion, respectively.

In addition to such quantitative differences between the approximate spin representations via truncated Taylor and Newton series, there is an important qualitative difference as far as the validity of the spin commutation relations (3.2) is concerned. In order to respect, e. g., rotational symmetries of the original Hamiltonian, we require for a proper approximation scheme that these commutation relations are satisfied within the subspace with at most $r < 2S$ HP bosons. The lowest-order term of the expansion ($r = 0$), corresponding to $\hat{S}^+ \mapsto \hat{S}_0^+ = \sqrt{2S}\,\hat{a}$ and $\hat{S}^- \mapsto \hat{S}_0^- = \sqrt{2S}\,\hat{a}^\dagger$, yields the commutator $[\hat{S}_0^+, \hat{S}_0^-] = 2S$, which is only correct when applied to a state with zero HP bosons.

---

[5]    For $S = 1/2$, the linear term in the Newton expansion $h_1(\hat{n})$ is twice as large as in $h_{\text{T},1}(\hat{n})$.

To guarantee the commutation relations in the larger Hilbert space of up to $r$ HP bosons, we truncate the Newton series of $h(\hat{n})$ in the definition of $\hat{S}^{\pm}$ after the term of order $\hat{n}^{(r)}$, $h(\hat{n}) \mapsto h_r(\hat{n})$. This gives for the resulting approximate spin operators $\hat{S}_r^{\pm}$

$$[\hat{S}_r^+, \hat{S}_r^-] = 2\hat{S}^z + \mathcal{O}(\hat{n}^{(r+1)}) \qquad \text{and} \qquad [\hat{S}^z, \hat{S}_r^{\pm}] = \pm\hat{S}_r^{\pm}, \tag{3.5}$$

which match the correct spin commutation relations up to order $\hat{n}^{(r)}$, i. e., they are exact within the Hilbert space of containing up to $r$ bosons. With increasing $r$, the Hilbert space in which the spin commutation relations are respected increases.

In contrast, when using the Taylor instead of the Newton expansion and truncating after the term of order $\hat{n}^r$, we find that the commutation relation $[\hat{S}_{T,r}^+, \hat{S}_{T,r}^-] = 2\hat{S}^z + \mathcal{O}(\hat{n})$ contains an error of order $\hat{n}$, irrespective of the order $r$ of the expansion. Increasing the order $r$ does, in this case, *not* increase the Hilbert space in which the commutation relations are fulfilled.

We illustrate this for $r = 1$. Making use of the Newton expansion, we find

$$[\hat{S}_1^+, \hat{S}_1^-] = 2\hat{S}^z - \left(3 - 12S[1 - h(1)]\right)\hat{n}^{(2)}, \tag{3.6}$$

in which the deviation from the exact commutation relation only matters when applied to states with at least two bosons. In contrast, using the Taylor expansion and then inserting $\hat{n}^2 = \hat{n}^{(2)} + \hat{n}$ leads to

$$[\hat{S}_{T,1}^+, \hat{S}_{T,1}^-] = 2\hat{S}^z + \frac{\hat{n}}{4S} + \frac{3}{8S}\hat{n}^{(2)} \tag{3.7}$$

that, because of the term $\hat{n}/(4S)$, deviates from the exact commutation relation already when applied to a state with one boson only.

The main virtue of the Newton expansion as compared to the Taylor expansion, however, is that the full series (2.10) can be truncated at the order $r = 2S$,

$$h(\hat{n}) = h_{2S}(\hat{n}) = \sum_{k=0}^{2S} \frac{\hat{a}^{\dagger k}\hat{a}^k}{k!} \underbrace{\sum_{l=0}^{k}(-1)^{k-l}\binom{k}{l}\sqrt{1 - \frac{l}{2S}}}_{H_k = \Delta_n^k h(0)}, \tag{3.8}$$

to *exactly* reproduce all spin matrix elements within the bosonic Hilbert space with at most $2S$ bosons. This includes the spin commutation relations as well as the relation $h(\hat{n})|2S\rangle = 0$ ensuring that the unphysical part of the boson Hilbert space with more than $2S$ bosons is unreachable by applying $\hat{S}^-$. In that sense, the finite sum (3.8) provides an exact spin representation.

## Discussion

The expansion (2.10) has already been proven by performing normal ordering by induction [11]. This result has also already been applied to the Holstein-Primakoff root $h(\hat{n})$ from (3.1b), together with the claim that a truncation of the series would connect the physical part of the Hilbert space to the unphysical part with more than $2S$ bosons [12]. This is, however, not true. In contrast, the finite sum (3.8) leaves the physical and the unphysical parts of the Hilbert space unconnected.

The observation that the truncation of the normal-ordered expansion of the Holstein-Primakoff representation at the order $2S$ provides an *exact* spin representation was recently published in [13]. Instead of performing a Taylor expansion, the authors made the *ansatz* to write $h(\hat{n})$ as a normal-ordered series in the form (2.10). Using techniques known from flow-equation approaches, they derived and solved differential equations to find iterative equations

for their[6] $Q_k$. The first few terms of the normal-ordered expansion has also been found by using the method of matching matrix elements [14].

What, to the best of our knowledge, has not been realized yet is that, in order to easily derive a compact formula for the coefficients $H_k$ in (3.8), finite-difference calculus provides a natural and elegant tool that is well adapted to the discreteness of the domain of definition of $f(\hat{n})$. Furthermore, it directly leads to closed and compact expressions instead of iterative equations [13] for the coefficients of the expansion.

We remark that, in addition to the Holstein-Primakoff representation, there are also other exact bosonic representations of quantum spins. The Dyson-Maleev representation uses different functions $h_{\pm}(\hat{n})$ for the operators $\hat{S}^+$ and $\hat{S}^-$. While the choice $h_+(\hat{n}) = 1 - \hat{n}/(2S)$ and $h_- = 1$ made in the original proposal [15–17] does still connect the physical and the unphysical part of the Hilbert space, the conjugated Dyson-Maleev representation [18], $h_+ = 1$ and $h_-(\hat{n}) = 1 - \hat{n}/(2S)$ corrects this flaw. Since no square root needs to be expanded, the (conjugated) Dyson-Maleev representation shares with the Newton expansion that a finite polynomial of the number operator is sufficient to represent the spin. This comes, however, with the drawback that the matrix representations of the operators $\hat{S}^+$ and $\hat{S}^-$ in the Fock base $\{|n\rangle\}$ are no longer Hermitian conjugates of each other. As a consequence, $\hat{S}^x$ and $\hat{S}^y$ are no longer represented by Hermitian matrices.

Finally, we remark that the appearance of boson states that do not correspond to spin states can be avoided by using a multi-valued transformation between the spin and the boson Hilbert space [19–22]. Also for this transformation, normal-ordered expansions can be performed.

## 3.2 Coherent states

While the eigenstates $|n\rangle$ of the occupation number operator $\hat{n}$ form a discrete base of the Fock space, it is also possible to express the Fock states in terms of coherent states $|\alpha\rangle$, defined by being eigenstates of the annihilation operator, $a|\alpha\rangle = \alpha|\alpha\rangle$. Since the spectrum of $a$ is continuous, the coherent states form a continuous, overcomplete base of the Fock space. For the textbook example of a harmonic oscillator, coherent states are known to mimic the classical equations of motions while minimizing the uncertainty product. This is also the reason why laser light in the classical limit is most properly described by coherent states [23].

The expectation value of an operator function $f(\hat{n})$ with respect to a coherent state $|\alpha\rangle$ is most easily evaluated when $f(\hat{n})$ is written in its normal-ordered form. While rearranging the Taylor series of $f(\hat{n})$ into a normal-ordered form involves cumbersome multiple applications of the commutator $[\hat{a}, \hat{a}^\dagger] = 1$, leading to Stirling numbers of the second kind, the Newton series of $f(\hat{n})$ is already automatically normal ordered. Therefore, by making use of

$$\langle\alpha|\hat{n}^{(k)}|\alpha\rangle = \langle\alpha|\hat{a}^{\dagger k}\hat{a}^k|\alpha\rangle = (\alpha^*\alpha)^k, \tag{3.9}$$

we can immediately express the expectation value of $f(\hat{n})$ with respect to the coherent state $|\alpha\rangle$ as the power series

$$\langle\alpha|f(\hat{n})|\alpha\rangle = \sum_{k=0}^{\infty} \frac{\Delta_n^k f(0)}{k!} (\alpha^*\alpha)^k. \tag{3.10}$$

The corresponding expression of the expectation value in terms of the coefficients of the Taylor series would acquire a much more complicated structure that is impractical for actual calculations.

The coherent state $|\alpha\rangle$ can be constructed by applying the displacement operator

$$\hat{D}(\alpha) = e^{\alpha\hat{a}^\dagger - \alpha^*\hat{a}} \tag{3.11}$$

---

[6]     The coefficients $Q_k$ in [13] are related to our $H_k$ via $Q_k = H_k/k!$.

onto the vacuum state $|0\rangle$. This can be shown by rewriting the displacement operator with the help of the Baker-Campell-Hausdorff formula in a normal-ordered form to derive the relation

$$|\alpha\rangle = \hat{D}(\alpha)|0\rangle = e^{-\frac{1}{2}\alpha^*\alpha}e^{\alpha\hat{a}^\dagger}e^{-\alpha^*\hat{a}}|0\rangle = e^{-\frac{1}{2}\alpha^*\alpha}\sum_{n=0}^{\infty}\frac{\alpha^n}{\sqrt{n!}}|n\rangle, \quad (3.12)$$

which immediately yields that $|\alpha\rangle$ is an eigenstate of $\hat{a}$ with eigenvalue $\alpha$.

Interpreting $\hat{D}(\alpha)$ as a unitary transformation, the expectation value $\langle\alpha|f(\hat{n})|\alpha\rangle$ with respect to the coherent state $|\alpha\rangle$ can be reexpressed as the *vacuum* expectation value of the *displaced* operator $\hat{D}^\dagger(\alpha)f(\hat{n})\hat{D}(\alpha)$. Therefore, we are interested in analyzing how the normal-ordered Newton series of $f(\hat{n})$ behaves under the displacement transformation. By making use of $\hat{D}^\dagger(\alpha)\hat{a}\hat{D}(\alpha) = \hat{a} + \alpha$ and $\hat{D}^\dagger(\alpha)\hat{a}^\dagger\hat{D}(\alpha) = \hat{a}^\dagger + \alpha^*$, which are easily proven by taking the derivatives $\partial_\alpha$ and $\partial_{\alpha^*}$, respectively, we end up with

$$\hat{D}^\dagger(\alpha)f(\hat{n})\hat{D}(\alpha) = \sum_{k=0}^{\infty}\frac{\Delta_n^k f(0)}{k!}(\hat{a}^\dagger + \alpha^*)^k(\hat{a} + \alpha)^k. \quad (3.13)$$

As a result, displacing the operator $f(\hat{n})$ is simply achieved by replacing $\hat{a} \mapsto \hat{a} + \alpha$ and $\hat{a}^\dagger \mapsto \hat{a}^\dagger + \alpha^*$ in the Newton series expansion of $f(\hat{n})$, which does not spoil the normal-ordered structure of the expression.

For a simple illustration, we mention the quantum analog of the phase space, namely the Husimi distribution (see e.g. [9]), for a harmonic oscillator (we again set $\hbar = 1$) with Hamiltonian $\hat{\mathcal{H}} = \omega(\hat{n} + n_0)$ in thermal equilibrium at inverse temperature $\beta = 1/k_B T$. By setting $f(\hat{n})$ in (3.13) to the Boltzmann operator $f(\hat{n}) = \frac{1}{Z}e^{-\beta\hat{\mathcal{H}}}$, we obtain the thermal density operator

$$\hat{\rho}_{\text{th}}(\alpha, \beta) = \frac{1}{Z}\hat{D}^\dagger(\alpha)e^{-\beta\hat{\mathcal{H}}}\hat{D}(\alpha) = -\sum_{k=0}^{\infty}\frac{(e^{-\beta\omega} - 1)^{k+1}}{k!}(\hat{a}^\dagger + \alpha^*)^k(\hat{a} + \alpha)^k, \quad (3.14)$$

with partition function $Z = \text{Tr}\,e^{-\beta\hat{\mathcal{H}}}$. The Husimi distribution is, then, nothing but the vacuum expectation value,

$$Q(\alpha, \beta) = \frac{1}{\pi}\langle 0|\hat{\rho}_{\text{th}}(\alpha, \beta)|0\rangle = \frac{1}{\pi}(1 - e^{-\beta\omega})e^{(e^{-\beta\omega} - 1)\alpha^*\alpha}, \quad (3.15)$$

where the real and imaginary part of $\alpha$ are the analogs of position and momentum in classical phase space. As discussed in [9], the Husimi distribution in terms of $\Re(\alpha)$ and $\Im(\alpha)$ is Gaussian as in the classical case, but with a larger width since quantum fluctuations add to the thermal ones. The use of the Newton series expansion helped us to derive the Husimi distribution in a simple and straightforward way without the cumbersome use of commutation relations when writing the Hamiltonian in a normal-ordered form. While for the harmonic oscillator, a compact expression for the Husimi function could be found, a series expansion of the Husimi function for any system with a Hamiltonian diagonal in $\hat{n}$ is easily obtained via its Newton series.

## 3.3 Photon statistics

The third example we study is taken from quantum optics. In this field, analyzing the statistics of the number of photons that result from probing a quantum-mechanical electromagnetic field with photon detectors is a major subject [23].

The statistical properties of the number $n$ of detected photons are contained in a probability distribution function. It is well known from probability theory that distribution functions can

be characterized in terms of moments $M_k$ of order $k$, which can be represented, using the moment generating function[7]

$$\mathcal{M}(z) = \sum_{k=0}^{\infty} M_k \frac{z^k}{k!}, \tag{3.16}$$

as the $k$-th derivative with respect to the real or complex auxiliary variable $z$ of $\mathcal{M}(z)$ at $z = 0$, i.e., $M_k = \partial_z^k \mathcal{M}(0)$. To get the corresponding *cumulant* $C_k$ of order $k$, one needs to take the logarithm before performing the derivatives, $C_k = \partial_z^k \ln \mathcal{M}(0)$. Remember that generating functions are always expanded in a Taylor series in $z$.

In the context of photon counting, the moment generating function is written as the quantum-mechanical expectation value

$$\mathcal{M}(z) = \langle \mathcal{G}(z; \hat{n}) \rangle \tag{3.17}$$

of some operator generating function $\mathcal{G}(z; \hat{n})$, which opens the question of what is the most natural form for $\mathcal{G}$. While $\mathcal{M}(z)$ is a real (or complex) function generating the moments $M_k$, the corresponding $\mathcal{G}(z; \hat{n})$ is a suitable operator function in Fock space that, expanded formally in $\hat{n}$, generates the powers $z^k$ in the series expansion (3.16).

The most obvious choice is the formal power series

$$\mathcal{G}_{\mathrm{r}}(z; \hat{n}) = \sum_{k=0}^{\infty} z^k \frac{\hat{n}^k}{k!} = e^{z\hat{n}}, \tag{3.18}$$

which corresponds to a Taylor expansion in $\hat{n}$ and yields the *raw* or *ordinary* moments

$$M_{\mathrm{r},k} = \langle \hat{n}^k \rangle. \tag{3.19}$$

However, as discussed in section 2, the proper expansion of a number operator function $f(\hat{n})$ in $\hat{n}$ is the Newton expansion. Adopting the form of equation (2.6a) yields

$$\mathcal{G}_{\mathrm{f}}(z; \hat{n}) = \sum_{k=0}^{\infty} z^k \frac{\hat{n}^{(k)}}{k!} = (1 + z)^{\hat{n}}, \tag{3.20}$$

as an alternative choice for the operator function $\mathcal{G}$. To prove the second equality in (3.20), we use (2.10) as well as the binomial theorem for $[(1 + z) - 1]^k$, such that the expansion coefficients of the Newton series of $(1 + z)^{\hat{n}}$ are given by

$$G_k = \sum_{l=0}^{k} (-1)^{k-l} \binom{k}{l} (1 + z)^l = z^k, \tag{3.21}$$

as required. The resulting moments

$$M_{\mathrm{f},k} = \langle \hat{n}^{(k)} \rangle = \langle :\hat{n}^k: \rangle, \tag{3.22}$$

generated by this second choice, $\mathcal{M}_{\mathrm{f}}(z) = \langle (1 + z)^{\hat{n}} \rangle$, are called the *factorial* moments of order $k$. They differ from the raw moments (3.19) by normal ordering of the photon creation and annihilation operators. The normal ordering also reflects the fact that each photon is destroyed upon detection [23].

The relation between the operator functions (3.18) and (3.20) can be summarized as the operator identity

$$\mathcal{G}_{\mathrm{f}}(z; \hat{n}) = (1 + z)^{\hat{n}} = :e^{z\hat{n}}: = :\mathcal{G}_{\mathrm{r}}(z; \hat{n}):, \tag{3.23}$$

---

[7] To be more precise, $\mathcal{M}$ is an *exponential* generating function, which involves a factor $1/k!$.

i. e., $\mathcal{G}_{\mathrm{f}}(z;\hat{n})$ is obtained from $\mathcal{G}_{\mathrm{r}}(z;\hat{n})$ by applying the normal-ordering operator. In the framework of coherent states, using (3.13) this identity translates to

$$\langle\alpha|\mathcal{G}_{\mathrm{f}}(z;\hat{n})|\alpha\rangle = \sum_{k=0}^{\infty}\frac{z^k}{k!}(\alpha^*\alpha)^k = e^{z\alpha^*\alpha} = \mathcal{G}_{\mathrm{r}}(z;\alpha^*\alpha). \qquad (3.24)$$

We remark that beyond the specific example discussed here in the context of photon statistics, the relation between operator-valued functions by applying the normal-ordering operator can, more generally, be derived with the help of an integral transformation between complex-valued functions that we introduce and analyze in section 4 of this paper.

**Discussion**

The identity $\mathcal{M}_{\mathrm{f}}(z) = \langle(1+z)^{\hat{n}}\rangle = \langle\,:e^{z\hat{n}}:\,\rangle$ has been proven in [23] by making use of the optical equivalence theorem [24, 25], which expresses the formal equivalence between expectations of normal-ordered operators in quantum optics and the corresponding $c$-number function in classical optics. By performing a Newton expansion of the operator function $\mathcal{G}_{\mathrm{f}}$, we were able to prove the stronger operator identity (3.23), without making any assumptions about the occupied states of the electromagnetic field.

We have seen that the factorial moments $M_{\mathrm{f},k} = \langle\hat{n}^{(k)}\rangle$ (and the corresponding cumulants $C_{\mathrm{f},k}$) naturally arise from the Newton expansion (3.20) of the operator function $\mathcal{G}_{\mathrm{f}}(z;\hat{n})$ entering the moment generating function $\mathcal{M}_{\mathrm{f}}(z) = \langle\mathcal{G}_{\mathrm{f}}(z;\hat{n})\rangle$. It is, of course, legitimate to also characterize the discrete photon statistics in terms of raw moments $M_{\mathrm{r},k} = \langle\hat{n}^k\rangle$ rather than the factorial ones. Since factorial moments can be expressed in terms of raw moments and vice versa, both descriptions contain the same information. Nevertheless, the use of factorial moments is more natural for *discrete* probability distributions. Above, we have seen that they are a consequence of the Newton expansion. The factorial moments are not only more natural, they are also superior over raw ones.

To illustrate this, we consider as a simple example an ensemble of $N$ independent photon sources, each emitting a photon with probability $p$. The resulting binomial distribution function of finding $n$ photons reads $P(n) = \binom{N}{n}p^n(1-p)^{N-n}$, and the factorial moment generating function becomes

$$\mathcal{M}_{\mathrm{f}}(z) = \sum_{n=0}^{N}\mathcal{G}_{\mathrm{f}}(z;n)P(n) = (1+pz)^N\,, \qquad (3.25a)$$

such that the factorial moments (3.22) are

$$M_{\mathrm{f},k} = N^{(k)}p^k\,, \qquad (3.25b)$$

which implies $M_{\mathrm{f},k} = 0$ for $k > N$, i. e., only a finite number of factorial moments are required to fully describe the photon distribution. In contrast, raw moment generating function is given by $\mathcal{M}_{\mathrm{r}}(z) = [1 + p(e^z - 1)]^N$ for this example and leads to an infinite number of quite complicated raw moments that grow exponentially, $M_{\mathrm{r},k} \simeq N^k p^N$, for $k \gg N$.

Beyond this specific example, it is quite straightforward to reconstruct any probability distribution $P(n)$ from the probability generating function $\mathcal{M}_{\mathrm{f}}(z-1)$ via

$$P(n) = \frac{1}{n!}\partial_z^n\mathcal{M}_{\mathrm{f}}(-1) = \frac{1}{n!}\sum_{k=0}^{\infty}M_{\mathrm{f},k}\frac{\partial_z^n z^k}{k!}\bigg|_{z=-1} = \frac{1}{n!}\sum_{k=0}^{\infty}\frac{(-1)^k}{k!}M_{\mathrm{f},n+k}\,. \qquad (3.26)$$

The denominator $k!$ guarantees a fast convergence for well-behaved factorial moments. In contrast, the relation between $P(n)$ and $M_{\mathrm{r},k}$ is not only more complicated, but it also bears

the problem of bad convergence since raw moments generically grow exponentially with the order $k$. This shows that raw moments and cumulants may be suited to characterize *continuous* probability distributions, but *discrete* probability distributions should rather be analyzed with the help of factorial moments and cumulants.

Not only photons but also electrons can be counted. The fact that they are fermions rather than bosons is irrelevant for the definition (not for the value) of the moments characterizing the probability function. Similar ideas as those for addressing photon detection in electromagnetic fields have been used to develop a theory of electron counting statistics for transport in nanostructures [26]. Quite surprisingly, most of the theoretical and experimental works on electron counting statistics have discussed the statistical properties in terms of raw moments and cumulants. This is despite the fact that electrons carry quantized charges and that, similar to the detected photons in quantum optics, once the transfer of an electron is measured it is out of the game. From the above discussion, it is obvious that one should rather use factorial moments and cumulants also for electron counting statistics. The virtue of factorial cumulants has only been realized later [27–29]. Their supremacy over raw cumulants in identifying, e. g., the nonequilibrium dynamics of spin relaxation in singly-charged quantum dots has been experimentally demonstrated recently [30].

## 4  Normal-order transform

In section 3.3, we have seen that the discreteness of the spectrum of $\hat{n}$ suggests to replace the operator function $\mathcal{G}_{\mathrm{r}}(z;\hat{n}) = e^{z\hat{n}}$, leading to raw moments, by the operator function $\mathcal{G}_{\mathrm{f}}(z;\hat{n}) = (1+z)^{\hat{n}}$, leading to factorial moments. One way to get from $e^{z\hat{n}}$ to $(1+z)^{\hat{n}}$ is to expand the former into a formal power series and then apply the normal-ordering operator $:\cdot:$, see (3.23). To put this procedure on a more formal ground, we introduce the transformation

$$f(x) \mapsto \tilde{f}(n) = \mathcal{N}_x[f(x)](n), \tag{4.1}$$

between (continuous) functions $f$ and $\tilde{f}$ that fulfills the operator identity

$$:f(\hat{n}): = \mathcal{N}[f](\hat{n}). \tag{4.2}$$

Since the transformation (4.1) is defined by a normal-ordering procedure, we call $\mathcal{N}[f]$ the *normal-order transform* of $f$. In the above example, the normal-order transform of $f(\hat{n}) = e^{z\hat{n}}$ is $\mathcal{N}[f](\hat{n}) = (1+z)^{\hat{n}}$. Our aim in this section is to find a general expression for the normal-order transform for an arbitrary function $f(x)$.

The connection (2.8) between normal ordering and factorial powers, $:\hat{n}^k: = \hat{n}^{(k)}$, implies that the powers $f(x) = x^k$ with $k \in \mathbb{N}_0$ have to be mapped onto the factorial powers $\mathcal{N}[f](n) = n^{(k)}$. As consequence, the Newton series of $\tilde{f}(n)$ must have the same coefficients as the Taylor series of $f(x)$, such that

$$F_k = \Delta_n^k \tilde{f}(0) = \partial_x^k f(0). \tag{4.3}$$

We can construct $\mathcal{N}$ by analytic continuation of the well-known integral representation of the Euler gamma function, valid for $n \in \mathbb{C}$, $\Re(n) < 0$,

$$\Gamma(-n) = (-1)^{n+1} \int_{-\infty}^{0} \mathrm{d}x\, e^x\, x^{-(n+1)}, \tag{4.4}$$

and find, for $k \in \mathbb{Z}$, the integral representation of the factorial power

$$n^{(k)} = (-1)^k \frac{\Gamma(k-n)}{\Gamma(-n)} = \frac{(-1)^{n+1}}{\Gamma(-n)} \int_{-\infty}^{0} \mathrm{d}x\, e^x\, x^{k-(n+1)} \tag{4.5}$$

as an integral transform of $x^k$. This holds both for positive and negative exponents of the factorial power, where the latter are defined via $n^{(-k)}(n+k)^{(k)} = 1$. In (4.5), we used the symmetry property $\Gamma(n)\Gamma(1-n)\sin(\pi n) = \pi$ of the gamma function. The analytic continuation to positive $n$ can be done by utilizing a Hankel contour along the negative axis in (4.4), with the result that the diverging contributions in the two integrals cancel for $k \in \mathbb{Z}$.

Applying the transform (4.5) term-wise to the Taylor series of $f(x)$ and interchanging sum and integral, we find the explicit expression

$$\tilde{f}(n) = \mathcal{N}[f](n) = \frac{1}{\Gamma(-n)} \int_{-\infty}^{0} \mathrm{d}x\, f(x)\, e^x\, (-x)^{-(n+1)}, \tag{4.6}$$

which can be used in, e.g., *Mathematica* to calculate the normal-order transform of many functions, see Table 1.

**Discussion**

The normal-order transform of $f(x)$ is directly related to the well-known Mellin transform $\mathcal{M}_x$ of $e^{-x}f(-x)$ according to

$$\mathcal{N}_x[f(x)](n) = \frac{1}{\Gamma(-n)} \mathcal{M}_{-x}[e^x f(x)](-n). \tag{4.7}$$

Under certain conditions [31], the Mellin transform is invertible, such that the inverse normal-order transform can also be given and reads

$$f(x) = \mathcal{N}_n^{-1}[\tilde{f}(n)](x) = e^{-x} \mathcal{M}_{-n}^{-1}[\Gamma(-n)\tilde{f}(n)](-x)$$
$$= \frac{e^{-x}}{2\pi \mathrm{i}} \int_{\mathcal{C}} \mathrm{d}n\, \tilde{f}(n)\, \Gamma(-n)\, (-x)^n, \tag{4.8}$$

where $\mathcal{C}$ is an appropriate contour in the complex plane. This inverse transform can be interpreted as a variation of the Nørlund–Rice integral, such that the normal-order transform together with its inverse resembles a summated version of the so-called Poisson-Mellin-Newton cycle [32]. Both Mellin transform and its inverse are tabulated and can be calculated with, e.g., *Mathematica*, see Table 1 for some examples.

Table 1: Normal-order transform $\tilde{f}(\hat{n})$ of selected functions $f(\hat{n})$, together with the common series coefficients $F_k$ of the respective Newton and Taylor series. We have the Bessel function of the first kind $J_0$, the Laguerre polynomials $L_n$, the exponential integral function $E_n$, the Hermite polynomials $H_n$, the generalized Riemann zeta function $\zeta$, the Bernoulli polynomials $B_k$, and the Lerch transcendent $\Phi$.

| $f(\hat{n}) = \mathcal{N}^{-1}[\tilde{f}](\hat{n})$ | $\tilde{f}(\hat{n}) = \mathcal{N}[f](\hat{n})$ | $F_k/k!$ | note |
|---|---|---|---|
| $\hat{n}^r$ | $\hat{n}^{(r)}$ | $\delta_{k,r}$ | equation (4.5) |
| $e^{z\hat{n}}$ | $(1+z)^{\hat{n}}$ | $z^k/k!$ | equation (3.23) |
| $J_0(2\sqrt{z\hat{n}})$ | $L_{\hat{n}}(z)$ | $(-z)^k/k!^2$ | see e.g. Ref. [33] |
| $\frac{1}{z-\hat{n}}$ | $e^z E_{-\hat{n}}(z)$ | $z^{-(k+1)}$ | geometric series |
| $e^{-z^2\hat{n}^2}$ | $z^{\hat{n}} H_{\hat{n}}(\frac{1}{2z})$ | $(-z)^{k/2}/(\frac{k}{2})!$ | $F_k = 0$ for odd $k$ |
| $e^{z\hat{n}}/(e^{y\hat{n}}-1)$ | $-y^{\hat{n}}\zeta(-\hat{n}, \frac{z+1}{y})$ | $y^k B_{k+1}(\frac{z}{y})/(k+1)!$ | $k \geq -1$, see (4.9) |
| $e^{z\hat{n}}/(e^{y\hat{n}}w \pm 1)$ | $\pm y^{\hat{n}}\Phi(\mp w, -\hat{n}, \frac{z+1}{y})$ | complicated | $k \geq -1$, see Ref. [34] |

Up to here, we only considered functions $f(x)$ that have a well-defined Taylor series around $x = 0$. We now demonstrate, that the normal-order transform can also deal with functions such as[8] $f(x) = 1/(e^x - 1)$, which has a simple pole at the origin, with residuum 1. Its normal-order transform $\tilde{f}(n) = -\zeta(-n)$ involves the famous Riemann zeta function $\zeta$. While $f(x)$ has to be expanded into a Laurent series starting with power $k = -1$, the transform $\tilde{f}(n)$ has a well-defined Newton expansion starting with factorial power $k = 0$. However, the expansion coefficients of the two series do not match each other. This putative inconsistency with (4.3) can be resolved by extending the notion of the Newton series to that of a *generalized* Newton series, by simply starting the expansion with a suitable factorial power $k < 0$. Here, this yields

$$f(x) = \frac{1}{e^x - 1} = \sum_{k=-1}^{\infty} \frac{B_{k+1}}{(k+1)!} x^k \quad \longleftrightarrow \quad \tilde{f}(n) = -\zeta(-n) = \sum_{k=-1}^{\infty} \frac{B_{k+1}}{(k+1)!} n^{(k)}, \qquad (4.9)$$

with the Bernoulli numbers $B_k$, and the expansion coefficients of both series are equal for all values of $k$ as required.

For the calculation of normal-ordered operator expansions, however, we have to eliminate the terms with $k < 0$ in $\tilde{f}(\hat{n})$. This is easily done by noting that, for $r > 0$,

$$n^{(-r)} = \frac{1}{(n+r)^{(r)}} = \frac{1}{r!} \sum_{k=0}^{\infty} \frac{(-1)^k}{\left(1 + \frac{k}{r}\right) k!} n^{(k)} \qquad (4.10)$$

can itself be expanded into a regular Newton series because, in contrast to $n^{-r}$, the closest simple pole of $n^{(-r)}$ is located at $n = -1$, well below the expansion point $n = 0$. The resulting resummed normal-ordered Newton series therefore becomes

$$\tilde{f}(\hat{n}) = -\zeta(-\hat{n}) = \sum_{k=0}^{\infty} \frac{B_{k+1} - (-1)^k}{(k+1)!} \hat{n}^{(k)}. \qquad (4.11)$$

Finally, we return to the context of coherent states, where the inverse normal-order transform (4.8) can be used to calculate the coherent state expectation value (3.10) of arbitrary number operator functions $\tilde{f}(\hat{n})$ in a closed form, as from (4.3)

$$\begin{aligned} \langle \alpha | \tilde{f}(\hat{n}) | \alpha \rangle &= \sum_{k=0}^{\infty} \frac{F_k}{k!} (\alpha^* \alpha)^k \\ &= f(\alpha^* \alpha) = \mathcal{N}^{-1}[\tilde{f}](\alpha^* \alpha), \end{aligned} \qquad (4.12)$$

see (3.15) and (3.24) for examples.

## 5  Conclusion

We have demonstrated that finite-difference calculus provides the natural basis for series expansions of functions $f(\hat{n})$ of occupation number operators $\hat{n}$. While the Taylor series, known from differential calculus, corresponds to an expansion in powers $\hat{n}^k$ of the number operator $\hat{n}$, the use of finite-difference calculus leads to Newton series. The Newton series of a number operator function corresponds to either normal ordering of the powers of $\hat{n}$ or, equivalently, to an expansion in terms of *factorial* powers $\hat{n}^{(k)}$ of $\hat{n}$. Newton series and factorial powers are superior to account for the discreteness of the spectrum of $f(\hat{n})$.

---

[8]    See bottom of Table 1 for a straightforward generalization to Bose-Einstein and Fermi-Dirac integrals, see, e. g., Ref. [34] and references therein.

We illustrated the usefulness of the Newton expansion with three examples. In the first one, the Newton expansion was applied to the Holstein-Primakoff representation of quantum spins. In contrast to the analogous Taylor expansion with infinitely many terms, the exact spin representation with the Newton expansion is already achieved for a *finite* sum of terms, with closed and compact expressions for the coefficients. Furthermore, for an approximative but systematic treatment of the spin operators, the Newton expansion is superior to the Taylor expansion: while the $r$-th partial sum of the Newton series for the spin representation renders the correct spin commutation relations within the subspace of up to $r$ Holstein-Primakoff bosons intact, the corresponding $r$-th partial sum of the Taylor expansion breaks the spin commutation relations when at least one Holstein-Primakoff boson is present.

In the second example, we demonstrated that the Newton series expansion is naturally connected to coherent states. In particular, expectation values of an operator function with respect to a coherent state are easily obtained from the function's Newton series. This provides, e. g., a convenient starting point for analyzing the Husimi distribution of a quantum-statistical system.

In the third example, we addressed the counting statistics of photons and electrons. We used the Newton expansion for a quick and easy derivation of the function that generates the relevant quantities characterizing the probability distribution of the counted photons or electrons. The discreteness of the counted objects suggests that one should use *factorial* moments (or cumulants) instead of raw ones. While this has been realized in the case of photon counting in quantum optics early on, it starts to be acknowledged only recently in the field of electron counting in transport through nanostructures.

Finally, we introduced the *normal-order transform* of an operator function, that is obtained by applying the normal-ordering operator on the formal power series of the function. We were able to represent both the normal-order transform as well as its inverse transform directly in terms of an integral transformation, and reexpressed it in terms of the well-known Mellin transform. This representation is related to the Poisson-Mellin-Newton cycle and avoids both the explicit evaluation of commutators as well as Stirling numbers that appear when expressing factorial powers through the usual ones.

## Acknowledgements

We thank Björn Sothmann and Eric Kleinherbers for useful discussions. We acknowledge financial support from the Deutsche Forschungsgemeinschaft (DFG, German Research Foundation) under Project-ID 278162697 – SFB 1242.

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
