# Peer review of "Newton series expansion of bosonic operator functions"

_SciPost Physics, doi:SciPost Phys. 10, 007 (2021)_

## Round 1 · Referee Report · Dirk Schuricht (Referee 1) · 2020-9-16

Report

The authors discuss the application of the Newton expansion to define operator-valued functions of the bosonic number operator. They briefly discuss the mathematical background and relation to the usual Taylor series. They present two specific examples: Holstein-Primakov representation of spins and photon statistics. In particular, they point out the advantages of using the Newton series over the usually applied Taylor series.

In my view, the article can be seen as a neat little note pointing out a fairly simple yet usually unknown mathematical method which can be applied in quantum mechanics to simplify and clarify operator-valued functions. As such I find the ideas of the note worthwhile publishing (maybe in SciPost Physics Core). However, I would appreciate some additions or remarks on further applications or generalisations, eg, on the applicability of the approach to operators that do not possess a spectrum of equally spaced real values, fermionic system, or systems with several degrees of freedom.

As a minor remark, I think the notation in (2) is somewhat imprecise regarding to which part of the expression the derivative is applied.

---

## Round 1 · Referee Report · Anonymous (Referee 2) · 2020-9-24

Strengths

The paper makes an interesting mathematical observation, which could be useful in several fields of theoretical physics.

Weaknesses

The paper provides two examples of how the Newton series can be useful, however, they are both somewhat abstract, and it would be nice to see a more concrete example, where this usefulness is evident.

Report

I can already see the report by the other reviewer, which I in many ways agree with. The authors are making an interesting observation, which might be of interest to other researchers working on quantum mechanical problems. It is perhaps hard to identify a very specific result in this manuscript, but it is indeed interesting to note how the Newton series has certain advantages over the standard Taylor expansion. As such, I believe that the paper should be published, as it may stimulate further theoretical developments in many different fields.

Requested changes

A minor suggestion would be not to use abbreviations such as "w.l.o.g.", which might not be known to all readers.

---

## Round 1 · Referee Report · Anonymous (Referee 3) · 2020-10-7

Strengths

Nice introduction/reminder of the Newton expansion for finite-difference calculus. The approach is naturally motivated for operators that take only discrete values. The expansion can be quite useful for square roots of operators, such as appear in the Holstein-Primakoff representation of spins. For this example, the Newton expansion terminates at a finite order, in contrast to the usual Taylor expansion used for spin wave computations. The Newton expansion thus reproduces all matrix elements within the bosonic Hilbert space exactly.

There is also a nice discussion of the application of the method to photon statistics, with an emphasis on the value of the “factorial moments” over the more conventional raw (or ordinary) moments.

Weaknesses

It would be nice if there were an example where a key physical result is correctly computed by this method, but not by the more conventional approaches.

Report

This paper presents a clear pedagogical explanation of the use of finite-difference calculus to accurately compute the expectation values of operator expansions that may formally have non-physical points of non-analyticity in the expansion, (e.g. sqrt operator around the origin).

I believe the acceptance criteria have been met.

---

## Round 3 · Author Response

Dear Editor, Dear Referees,

we thank all three referees very much for their positive assessment and their useful comments. Two of the referees ask for more examples and illustrations of the supremacy of the Newton over the Taylor expansion. Motivated and stimulated by these requests, we substantially enhanced our manuscript by adding several new aspects that show the usefulness of the Newton expansion and clarify its properties and connections to other fields in physics. The changes are detailed in the section "List of changes" below.
In conclusion, we are sure that we have substantially enhanced our manuscript, the additional text adds up to 6 pages in the draft. We have included more applications and more connections of the Newton expansion scheme to different fields in physics. Therefore, we are confident that our manuscript is now eligible for publication in SciPost Physics.

Yours sincerely,
Jürgen König
Alfred Hucht

---

## Round 3 · List of Changes

We substantially enhanced our manuscript v1 by adding several new aspects that show the usefulness of the Newton expansion and clarify its properties and connections to other fields in physics. To be more specific:

- As a new application, we have added a complete section 3.2 demonstrating that the Newton series expansion is naturally connected to coherent states. Expectation values of an operator function with respect to a coherent state are easily obtained from the function's Newton series, which provides, e.g., a convenient starting point for analyzing the Husimi distribution of a quantum-statistical system.

- In section 3.1 on the Holstein-Primakoff representation of spins, we have added a discussion illustrating that the Newton expansion allows for an approximative but systematic treatment of the spin operators in the sense that the r-th partial sum of the Newton series for the spin representation yields the correct spin commutation relations within the subspace of up to r Holstein-Primakoff bosons. In contrast, the corresponding r-th partial sum of the Taylor expansion already breaks the spin commutation relations when at least one Holstein-Primakoff boson is present.

- In section 3.3 on photon statistics, we have included a new paragraph that demonstrates more explicitly the supremacy of factorial over raw moments. We show that the expansion of a discrete probability function in terms of factorial moments converges fast, while the corresponding expansion in terms of raw moments bears the problem of bad convergence since raw moments generically grow exponentially with the order k.

- We have added a full subsection 2.4 that clarifies the connection between finite-difference calculus and commutators.

- We have added the new section 4, in which we introduced the "normal-order transform" of an operator function, which is obtained by applying the normal-ordering operator on the formal power series of the function. We were able to represent both the normal-order transform as well as its inverse transform directly in terms of an integral transformation, and reexpressed it in terms of the well-known Mellin transform.

- We have added six new references [4,5,6,31,32,33], updated Ref. [13], and have modified sentences all over the manuscript to improve the presentation. As requested by one of the referees, we now avoid abbreviations.

- Finally, we have slightly restructured the outline of the manuscript to account for the newly included material.

In conclusion, we have substantially enhanced our manuscript (by 6 pages). We have included more applications and more connections of the Newton expansion scheme to different fields in physics.

You are currently on this page

Resubmission 2008.11139v3 on 2 December 2020

---

## Editorial Decision

published